# Neuraminidase Antibody Response to Homologous and Drifted Influenza A Viruses After Immunization with Seasonal Influenza Vaccines

**DOI:** 10.3390/vaccines12121334

**Published:** 2024-11-27

**Authors:** Yulia Desheva, Maria Sergeeva, Polina Kudar, Andrey Rekstin, Ekaterina Romanovskaya-Romanko, Vera Krivitskaya, Kira Kudria, Ekaterina Bazhenova, Ekaterina Stepanova, Evelina Krylova, Maria Kurpiaeva, Dmitry Lioznov, Marina Stukova, Irina Kiseleva

**Affiliations:** 1FSBSI ‘Institute of Experimental Medicine’, 197022 Saint Petersburg, Russia; polina6226@mail.ru (P.K.); arekstin@yandex.ru (A.R.); sonya.01.08@mail.ru (E.B.); fedorova.iem@gmail.com (E.S.); krylova.evelina.03@mail.ru (E.K.); mariakurpyaeva@gmail.com (M.K.); irina.v.kiseleva@mail.ru (I.K.); 2Smorodintsev Research Institute of Influenza, Ministry of Health of the Russian Federation, 197022 Saint Petersburg, Russia; mari.v.sergeeva@gmail.com (M.S.); romromka@yandex.ru (E.R.-R.); vera.krivitskaya@influenza.spb.ru (V.K.); kira336@yandex.ru (K.K.); dlioznov@yandex.ru (D.L.); marina.stukova@influenza.spb.ru (M.S.)

**Keywords:** influenza, vaccines, antibodies, neuraminidase inhibition, persistence, cross-reaction

## Abstract

Background/Objectives: Humoral immunity directed against neuraminidase (NA) of the influenza virus may soften the severity of infection caused by new antigenic variants of the influenza viruses. Evaluation of NA-inhibiting (NI) antibodies in combination with antibodies to hemagglutinin (HA) may enhance research on the antibody response to influenza vaccines. Methods: The study examined 64 pairs of serum samples from patients vaccinated with seasonal inactivated trivalent influenza vaccines (IIVs) in 2018 according to the formula recommended by the World Health Organization (WHO) for the 2018–2019 flu season. Antibodies against drift influenza viruses A/Guangdong-Maonan/SWL1536/2019(H1N1)pdm09 and A/Brisbane/34/2018(H3N2) were studied before vaccination and 21 days after vaccination. To assess NI antibodies, we used an enzyme-linked lectin assay (ELLA) with pairs of reassortant viruses A/H6N1 and A/H6N2. Anti-HA antibodies were detected using a hemagglutination inhibition (HI) test. The microneutralization (MN) test was performed in the MDCK cell line using viruses A/H6N1 and A/H6N2. Results: Seasonal IIVs induce a significant immune response of NI antibodies against influenza A/H1N1pdm09 and A/H3N2 viruses. A significantly reduced ‘herd’ immunity to drift influenza A/H1N1pdm09 and A/H3N2 viruses was shown, compared with previously circulating strains. This reduction was most pronounced in strains possessing neuraminidase N2. Seasonal IIVs caused an increase in antibodies against homologous and drifted viruses; however, an increase in antibodies to drifting viruses was observed more often among older patients. The level of NI antibodies for later A/H1N1pdm09 virus in response to IIVs was statistically significantly lower among younger people. After IIV vaccination, the percentage of individuals with HI antibody levels ≥ 1:40 and NI antibody levels ≥ 1:20 was 32.8% for drift A/H1N1pdm09 virus and 17.2% for drift A/H3N2 virus. Antisera containing HI and NI antibodies exhibited neutralizing properties in vitro against viruses with unrelated HA of the H6 subtype. Conclusions: Drift A/H1N1pdm09 and A/H3N2 viruses demonstrated significantly lower reactivity to HI and NI antibodies against early influenza viruses. In response to seasonal IIVs, the level of seroprotection has increased, including against drift influenza A viruses, but protective antibody levels against A/H1N1pdm09 have risen to a greater extent. A reduced immune response to the N1 protein of the A/H1N1pdm09 drift virus was obtained in individuals under 60 years of age. Based on our findings, it is hypothesized that in the cases of a HA mismatch, vaccination against N1-containing influenza viruses may be necessary for individuals under 60, while broader population-level vaccination against N2-containing viruses may be required.

## 1. Introduction

The global incidence of influenza is estimated to be between 27 and 54 million people annually and between 300,000 and 650,000 hospitalizations (https://www.who.int/news/item/13-12-2017-up-to-650-000-people-die-of-respiratory-diseases-linked-to-seasonal-flu-each-year (accessed on 16 November 2024)). Numerous studies have demonstrated that vaccination can reduce the risk of seasonal influenza infection by 40–60% if circulating strains are well matched to the vaccine strains [1,2]. The main antigens of the influenza virus are surface glycosylated proteins, notably hemagglutinin (HA) and neuraminidase (NA). HA is the principal antigenic component of the influenza virus [3], and the primary marker for evaluating humoral immunity following infection or vaccination [4]. The strain composition of influenza vaccines has to be updated almost annually due to the high variability of HA of influenza viruses [5]. In cases of genetic drift or antigenic shift among circulating strains, vaccine-induced hemagglutination inhibition (HI) antibodies may exhibit reduced cross-neutralization efficacy [6]. Therefore, it is critical to consider immune responses targeting the second major antigenic component, NA, when assessing susceptibility post-vaccination. NA should be regarded as a key target in future influenza vaccine platforms [7,8]. Anti-NA antibodies reduce the severity of influenza infection, prevent secondary complications, and effectively limit viral transmission [9]. In adults, pre-existing antibodies to NA have been associated with reduced viral shedding and shortened illness duration during natural A(H1N1)pdm09 influenza infection [10]. Investigating anti-NA antibodies may contribute to the study of herd immunity against emerging influenza strains and help evaluate the protective efficacy of seasonal vaccines, especially when there is a mismatch between epidemic and vaccine strains [11].

Despite the acknowledged importance of NA-specific antibodies in the protection against influenza, the NA content in seasonal influenza vaccines is neither regulated nor assessed by manufacturers [12], and the threshold level of NA inhibitory antibodies conferring protection is not clearly defined [13]. It has long been established that an HI antibody titer > 1:40 is associated with a 50% reduction in the risk of influenza infection [14]. Memoli et al. demonstrated, using the intranasal challenge of healthy volunteers, that HI titers >1:40 were protective against mild to moderate influenza infection but did not entirely prevent symptom occurrence. While the HI antibody titers were correlated with some reduction in disease severity scores, baseline NA inhibitory antibody titers showed a stronger correlation with severity scores and had a greater impact on recovery than baseline HI titers [7].

Antigenic changes in the influenza virus’s surface glycoproteins enable the evasion of pre-existing humoral immunity [6]. Antigenic drift is observed not only in HA but also in NA [15,16]. Notably, the antigenic drift of NA may occur independently or even at a different rate compared with HA [15]. For instance, from 2010 to 2016, influenza vaccines contained the A/California/07/09(H1N1)pdm09-like virus, reflecting a slower antigenic drift in the HA of A/H1N1pdm09 viruses. However, human monoclonal antibodies identified antigenic sites in the NA of A/H1N1pdm09 that underwent changes shortly after the emergence of the new pandemic virus [15].

Since 2019, significant antigenic drift in N1 neuraminidase has been reported [16]. It was found that the NA of the circulating 2019 strain acquired a significant number of mutations in the head domain, particularly the loss of the N-linked glycosylation site at 245 position (due to S245N and S247T mutations). In relation to A/H3N2 viruses, the emergence of an N-linked glycosylation site at position 245 (due to S245N and S247T mutations) and the P468H substitution have led to significant antigenic changes in N2 neuraminidase since 2016 [17]. In March 2019, the World Health Organization (WHO) recommended replacing the A/H3N2 component of influenza vaccines for the Northern Hemisphere for the 2018–2019 epidemic season with viruses similar to A/Kansas/14/2017 (H3N2) [18]. As with HA, genetic changes in NA do not always translate to antigenic alterations. A single amino acid substitution in NA of A/Brisbane/59/2007 was shown to reduce inhibition by polyclonal antibodies targeting earlier strains. These data suggest the importance of NA inhibition estimation to detect antigenic drift in the presence of sequence changes in NA [15].

Given the limited data on the efficacy of influenza vaccines in cases of mismatch between vaccine and epidemic strains, the aim of this study was to investigate the cross-reactivity and functional properties of NA inhibitory antibodies elicited by immunization with seasonal influenza vaccines.

## 2. Materials and Methods

### 2.1. Surveyed Contingents and Samples

Blood serum samples were obtained as part of a serological study [19] involving healthy volunteers over 18 years of age who had no contraindications to vaccination and who signed written informed consent. Immunization was performed in an open mode as previously described in a specialized clinic of the A.A. Smorodintsev Research Institute of Influenza. This study used blood samples obtained before vaccination and on the 21st day after vaccination. The study was approved by the Local Ethics Committee of the A.A. Smorodintsev Research Institute of Influenza, protocol No. 131 dated 10 October 2018. The study used three sеasonal trivalent inactivated influenza vaccines (IIV) with the composition of vaccine strains recommended by the WHO for the Northern Hemisphere in the 2018–2019 flu season: A/Michigаn/45/2015(H1N1)pdm09-like virus; A/Singapore/INFIMH-16-0019/2016 (H3N2)-like virus; and B/Colorado/06/2017 (B/Victoria/2/87 linеage)-like virus [14]. All vaccines were produced according to the Russian Pharmacopeia from purified candidate viruses grown in embryonated chicken eggs. The content of HA in the vaccine formulations was estimated using the single radial immunodiffusion method. The split vaccine “Ultrix” (“FORT”, Moscow, Russia) contained 15 micrograms of HA from each strain in a dose of 0.5 mL [20]. The subunit vaccine “Grippol Plus” (Petrovax Pharm, Moscow, Russia) contained 5 μg HA of each strain (antigens manufactured by Abbott Biologicals BV, Olst, the Netherlands) and 500 micrograms of Polyoxidonium^®^ adjuvant in a dose of 0.5 mL [21]. Another subunit vaccine, “Sovigripp” (NPO “Mikrogen”, Republic of Bashkortostan, Russia) contained 5 μg of HA from influenza strains A(H1N1)pdm09 and A(H3N2), 11 μg of influenza B strain, and 500 μg of Sovidon^®^ adjuvant in a dose of 0.5 mL [22]. There is no specific information regarding the amount of NA in any of the vaccines used in our study. The inclusion of NA in the Grippol vaccine is mentioned in the product information, which can be found on the manufacturer’s website (https://grippol.ru/en/grippol-kvadrivalent/instruction/, accessed on 18 November 2024).

After receiving permission from the Local Ethics Committee at the Federal State Budgetary Scientific Institution “IEM” No. 3/23 dated 20 September 2023, the clinical samples were transferred in an anonymized form to researchers.

### 2.2. Influenza Viruses

Influenza viruses A/Michigan/45/2015(H1N1)pdm09; A/Singapore/INFIMH-16-0019/2016 (H3N2), A/Guangdong-Maonan/SWL1536/2019(H1N1)pdm09 and A/Brisbane/34/2018 (H3N2) were used as antigens to perform HI.

We used H6Nx reassortant viruses containing NA of recent A/H1N1pdm09 and A/H3N2 viruses (Table 1) to measure NI titers in enzyme-linked lectin assay (ELLA).

Viruses were purified and concentrated by ultracentrifugation as previously described [19].

### 2.3. The Enzyme-Linked Lectin Assay (ELLA)

For the ELLA setup described previously [19], 96-well flat-bottomed ELISA plates with high binding were used. A total of 150 μL of fetal calf serum fetuin solution at a concentration of 50 μg/mL in 0.1 M carbonate/bicarbonate buffer (pH = 9.5–9.7) was added to the wells. Substrate adsorption occurred at a temperature of +4 °C overnight. Before immediate use, the plates were washed twice with sterile PBS (pH~7.4). The working dilution of diagnostic viruses A/H6N1 and A/H6N2 was prepared in 0.01 M PBS containing BSA at a concentration of 10 mg/mL. The 65 μL of the working dilution of the virus was added to each well of the plate with diluted sera. For control, control samples were placed in each plate: negative control, 130 µL of 0.01 M PBS with a BSA content of 10 mg/mL, and positive control, a mixture of 65 µL of the working dilution of the virus with an equivalent volume of 0.01 M PBS with a BSA concentration of 10 mg/mL. The result of the enzymatic reaction was obtained in optical density (OD 450 nm) data determined for a series of dilutions of each blood serum. TMB was used as a dye. The OD set was used to plot the graph of the dependency of the residual enzyme activity of NA in the presence of anti-NA antibodies, calculated as a function of serum dilution.

Serum antibody titer was determined as reciprocal dilution of the sample giving 50% NA inhibition. Results are expressed as Log2of reciprocal final dilution.

### 2.4. Hemagglutination Inhibition Test (HI)

HI was performed as described previously World Health Organization. Manual for the Laboratory Diagnosis and Virological Surveillance of Influenza. 2011. Available online: https://apps.who.int/iris/handle/10665/44518 (accessed on 13 November 2024). In brief, influenza antigens bearing the same hemagglutinin as in the vaccine strains were used. Sera were treated with a receptor-destroying enzyme (RDE, Denka Seiken Co., Tokyo, Japan) according to the manufacturer’s instructions. Each blood serum sample was 2-fold serially diluted in 96-well polypropylene U-bottom plates starting from 1:10 and mixed with 4 HAU of influenza antigen. After 1 h of incubation, 0.5% suspension of chicken red blood cells was added. Antibody seroconversion was identified as a fourfold or greater increase in the HI antibody titer compared with baseline.

### 2.5. Microneutralization Reaction (MN)

For the MN setup, a monolayer of MDCK cells was grown in 96-well flat-bottom polystyrene plates for adherent cultures (Sarstedt, Nümbrecht, Germany). After removing the maintenance medium and washing the plates twice with phosphate-buffered saline, 50 μL of falling two-fold dilutions (starting with a 1:10 dilution) of blood serum samples in DMEM containing trypsin TPCK at a concentration of 2 μg/mL were added. Then, 50 μL of a standard dose of virus (200 TCID_50_/0.05 mL = 100 TCID_50_/0.1 mL), diluted to the indicated concentration with the same medium, was added to each well. The plates were incubated in a CO_2_-fed thermostat at 34 °C for 72 h. Inhibition of virus reproduction was determined by a hemagglutination assay with 0.75% suspension of chicken erythrocytes.

### 2.6. Statistical Processing of Results

The results were processed using the statistical package “GraphPad Prism 6”. The following descriptive statistics were used to describe the data obtained: arithmetic mean (M), and standard deviation (σ). When comparing samples in case of failure to meet the assumptions of normal distribution of the dependent variable within each group and homogeneity of variance, nonparametric criteria were used (Mann–Whitney, Wilcoxon signed ranks, Friedman rank analysis of variance, and Kruskal–Wallis rank analysis of variance). For nominal data, Fisher’s exact test was used. The null hypotheses tested by the criteria were rejected at *p* < 0.05.

## 3. Results

### 3.1. Genetic Analysis of NA Subtypes N1 and N2 and Enzymatic Activity

Seasonal IIVs used in this study included vaccine strains of influenza A viruses recommended by WHO for the Northern Hemisphere in the 2018–2019 season: A/Michigan/45/2015 (H1N1)pdm09-like virus; A/Singapore/INFIMH-16-0019/2016 (H3N2)-like virus. To study immunity to drift variants, we used A/Guangdong-Maonan/SWL1536/2019 (reference virus recommended for use in the 2020–2021 Northern Hemisphere vaccine) and influenza virus A/Brisbane/34/2018 (A/Kansas/14/2017-like (recommended for use in the 2019–2020 Northern Hemisphere vaccine) [23].

A total of 20 substitutions (4.3% divergence) were identified in the NA amino-acid sequence (positions 1–469) of the circulating 2019 strain A/Guangdong-Maonan/SWL1536/2019(H1N1)pdm09 compared with the previously circulating strain A/South Africa/3626/2013(H1N1)pdm09. Several mutations were detected in the head domain (Figure 1А). As reported previously, one of the primary drivers of the antigenic changes in the NA A/Guangdong-Maonan/SWL1536/2019(H1N1)pdm09 is the loss of the N-linked glycosylation site at Asn386, which can significantly affect the antigenicity of N1 [24]. Another antigenic site, which includes residues 270 and 314, is located on the opposite side of the NA monomer in a region recognized by both mouse and human NI antibodies. Although the two sites (386 and 270/314) may represent independent NI epitopes, it is possible that substitutions at positions 270 and 314 could influence the distal epitope masked by the glycan at Asn386 [16]. The T188I mutation is particularly important for altering the binding affinity of immune antisera [25].

Substitutions T188I, M314I, and K389I are linked to moderate-to-high levels of resistance to neuraminidase inhibitors [26,27].

Structural analysis showed that the amino acid change from valine to isoleucine (V321I) distorts the hydrophobic pockets and affects residues within the active site of NA. This mutation may destabilize a nearby loop containing the oseltamivir-interacting residue R368, suggesting that these residues could serve as potential markers for reduced susceptibility to oseltamivir [28].

The E432K mutation is associated with antigenic drift/escape and moderate resistance to neuraminidase inhibitors, including oseltamivir, zanamivir, and peramivir [29]. The D449N mutation has been implicated in viral oligomerization, host protein interaction, and small ligand binding, as predicted by FluSurver (A*STAR Bioinformatics Institute, Singapore). These alterations may lead to conformation and functional changes in the NA of A(H1N1)pdm09.

In the A/H3N2 lineage, strains A/Hong Kong/4801/2014(H3N2) and A/Brisbane/34/2018(H3N2) differed by 15 amino acid substitutions (3.2% divergence) in their NA sequences, with 9 of the most significant changes occurring in the head domain outside the active site (Figure 1B). Substitutions V149A, Y155H, S247T, and T392I are associated with potential resistance to NA inhibitors [30,31,32]. The E344K mutation has been previously reported to alter antigenic properties, affecting the binding of monoclonal antibodies [33]. The S245N substitution likely creates a new glycosylation site at positions 245–247. The glycan at this site, adjacent to the enzyme’s active site, may sterically hinder the binding of NA-inhibitory antibodies while also reducing viral replication in vitro in human nasal epithelial cells [34]. These genetic mutations in N2 may, therefore, lead to significant structural and functional changes in the corresponding protein.

In assays using the high-molecular-weight substrate fetuin, the H6N1/19 virus containing NA A/Guangdong-Maonan/SWL1536/2019 (H1N1)pdm09 exhibited slightly lower enzymatic activity compared with the H6N1/13 virus containing NA A/South Africa/3626/2013 in the linear range of the titration curve (Figure 1C). Similarly, the H6N2/18 virus containing NA from A/Brisbane/34/2018(H3N2) showed significantly lower enzymatic activity compared with the H6N2/14 virus containing the NA of the A/Hong Kong/4801/2014 (H3N2) (Figure 1C).

These findings suggest that the acquired mutations in NA of A/H1N1pdm09 did not substantially impact N1 enzymatic activity, whereas the activity of the newly emerged N2 was markedly reduced.

### 3.2. Antibody Responses to A/H1N1pdm09 and A/H3N2 Viruses in Human Sera Pre- and Post-Immunization with Seasonal IIVs (2018–2019 Formulation)

Figure 2A illustrates the analysis of antibodies against vaccine and drift influenza viruses in the serum of 64 patients of various ages. Blood samples were collected prior to immunization and 21 days after vaccine administration. The results demonstrated that vaccination with seasonal IIVs increased antibody titers to homologous and drift viruses, with a more pronounced response against drift viruses in older patients.

Sera analysis prior to vaccination showed a statistically significant decrease in NA inhibition for the H6N1/19 virus compared with the H6N1/13 virus (Figure 2B). For A/H3N2 viruses, despite a significant decrease in the enzymatic activity of NA in the A/Brisbane/34/2018(H3N2) strain compared with the A/Hong Kong/2014/8296(H3N2) virus, a notable change in the inhibitory activity of antibodies specific to the A/Hong Kong/2014/8296(H3N2) against A/Brisbane/34/2018 (H3N2) was observed in the same serum samples. Statistically significant differences in NI antibody levels to N2 were noted between older and younger adult patients, indicating a markedly reduced ‘herd’ immunity to drift influenza viruses A/H1N1pdm09 and A/H3N2 compared with earlier strains.

Seroconversions to influenza A virus antigens were defined as a fourfold increase in HI antibody titers and a twofold increase in NI antibody titers. As shown in Figure 2C, seroconversion rates were lower in the younger cohort. Interestingly, over half of the seroconversions to new strains coincided with those to earlier strains; however, a subset of increased antibody responses to new viruses was independent of the presence of responses to previous antigenic variants (Figure 2C).

Figure 3 displays significant increases in antibody titers following immunization using seasonal influenza vaccines, with these antibodies reacting to both earlier and more recent variants of A/H1N1pdm09 and A/H3N2 viruses. In all cases, the titers of antibodies against HA and NA for recently emerged viruses were significantly lower both before vaccination (S1 on the abscissa axis) and 21 days after vaccination (S2 on the abscissa axis).

We did not compare levels of antibodies to HA and NA because these values were obtained by different methods. However, Figure 3 shows that the mean titers of HI antibodies were almost always about two times lower than those of NI antibodies. Given that there is no precise definition for protective titers for NI antibodies, the clinical significance of these values is still unclear.

The levels of immunoprotection, expressed as the proportion of individuals with HI antibody titers ≥ 1:40 and NI antibody titers ≥ 1:20 are presented in Figure 4 and Figure 5. Figure 4 indicates the percentage of individuals with antibody titers 21 days post-vaccination to either HA (blue bars), NA only (red bars), or both HA and NA. Protective levels of either HA or NA were achieved by 85.9% and 90.6% of patients for A/H1N1pdm09 and A/H3N2 viruses, respectively (Figure 4A). However, for later variants, only 32.8% reached protective titers against A/H1N1pdm09 virus and just 17.2% against A/H3N2. Antibodies targeting only NA without concurrent HI antibody formation were detected in only 3–6% of cases for A/H1N1pdm09 viruses and for early A/H3N2 virus (Figure 4A,B), and no antibodies that respond exclusively to the NA were found for A/Brisbane/34/2018(H3N2) virus (Figure 4B).

The numbers of persons and percentages of seroprotection in age groups are given in Appendix A. The proportions of individuals with protective levels of antibodies to antigens of new viruses were generally higher in older patients both before and after vaccination although the differences were not statistically significant (Figure 5). Only for the vaccine virus A/H1N1pdm09, HI antibody titers were slightly higher in younger patients before vaccination, but after vaccination, the proportion of seroprotection became the same in both age groups (Figure 5). At the same time, the level of seroprotection against NA of the latter A/H1N1pdm09 virus after IIV vaccination was statistically significantly lower among the group of patients under 60 years old.

We investigated the neutralizing antibodies to H6Nx viruses before and after vaccination, as well as the degree of association of neutralizing antibodies with HI and NI antibodies within the same serum samples collected before and after vaccination. Vaccination with seasonal IIVs resulted in a statistically significant increase in neutralizing antibodies against viruses with an unrelated HA subtype, H6 (Figure 6A,C). MN antibodies to reassortant H6N1/19 virus, which contains NA from A/Guangdong-Maonan/SWL1536/2019(H1N1)pdm09, showed a medium positive correlation with HI antibodies (*p* = 0.002, Figure 6B). Conversely, neutralizing antibodies to the reassortant H6N2/18 virus containing NA from A/Brisbane/34/2018(H3N2) virus medium positively correlated with antibodies detected via ELLA (*p* < 0.01, Figure 6D). Thus, antisera with HI and NI antibodies exhibit neutralizing properties in vitro against viruses bearing an unrelated HA subtype, H6.

### 3.3. Antibodies to A/H1N1pdm09 and A/H3N2 Viruses in Human Sera Before and After Immunization with Either a Split or Subunit IIV

It was shown that the increase in HA antibodies to the drifted influenza A/H1N1pdm09 and A/H3N2 influenza viruses after immunization with the split vaccine was insignificant, while the subunit vaccine caused a significant increase in antibodies to HA and NA of the drifted virus (Figure 7A,С). At the same time, there were statistically significant increases in antibodies to HA from drifted viruses in response to both the split and subunit vaccines (Figure 7B,D).

Thus, acquired mutations in NA resulted in decreased binding to immune antisera obtained through immunization with earlier strains. It was shown that the titers of HI antibodies to the A/Michigan/45/2015(H1N1)pdm09 virus and NI antibodies to the A/South Africa/3626/2013 (H1N1)pdm09 strain in patients’ sera before vaccination were significantly higher than the titers of antibodies to HA and NA antibodies to the A/Guangdong-Maonan/SWL1536/2019 (H1N1)pdm09 virus. Significant increases in average antibody titers were observed for the A/Guangdong-Maonan/SWL1536/2019 (H1N1)pdm09 strain, rather than just earlier viruses. Thus, seasonal IIVs based on previous strains induce cross-reactive antibodies against later strains. Neutralizing antibodies significantly increased for reassortant H6N1/19 and H6N2/18 strains, including NA of the A/Guangdong-Maonan/SWL1536/2019(H1N1)pdm09 and to A/Brisbane/34/2018(H3N2) viruses, respectively, in the sera from patients vaccinated with influenza IIVs based on previous strains.

## 4. Discussion

Antibodies against influenza surface antigens are key determinants of human susceptibility to influenza infection. Antigenic changes in the HA protein can reduce vaccine efficacy and prevent recognition by pre-existing antibodies. Analysis of influenza viruses, including those from the 2019–2020 season, indicates continuous evolution, with each season yielding new genetic variants of A(H1N1)pdm09 and A(H3N2) influenza viruses that exhibit HA gene changes compared with the vaccine strains [35]. Changes in NA protein can enable influenza viruses to evade neutralizing antibodies generated from prior infections or vaccinations [36]. However, the impact of NA drift on vaccine effectiveness tends to be less pronounced than that of HA drift. In addition, а major advantage of NA is its separation from HA evolution and its ability to induce cross-protection. NA antibodies often display broad cross-reactivity within a subtype [37,38], suggesting that these antibodies may be used to predict ‘herd’ immunity against newly emerging influenza viruses and assess potential protection against seasonal influenza viruses, even when there is a mismatch in the HA antigen [11].

Health authorities such as the Centers for Disease Control and Prevention (CDC) and WHO recommend annual influenza vaccination as the most effective strategy to achieve high seroprotection rates within a community and minimize the impact of influenza. For influenza vaccines, seroprotection is commonly defined by an HI titer of 1:40 or higher. In general, studies have reported post-vaccination seroprotection rates in adults ranging from 70% to 90% [39], although these rates can vary depending on age, health status, vaccination history, and the match between vaccine and circulating strains. In our study, seroprotection rates after seasonal IIVs, as determined by HI antibodies, were 80% for recent strains and up to 33% for drifted strains.

Anti-NA titers in humans have been shown to independently correlate with protection, regardless of HI antibody levels [40]. Since HI acts as a signal peptide, neutralizing antibodies target HA in the early stages of infection, preventing the virus from attaching to cellular receptors [41]. NA, however, plays a critical role in the viral life cycle by facilitating the release of progeny viruses from infected host cells. There is evidence that аnti-NA immunity reduces virus shedding and infectivity in humans [10,42]. A recent household transmission study showed that higher pre-existing antibody levels to the HA head, HA stalk, and NA were associated with reduced susceptibility to infection. Notably, only anti-NA antibodies were linked to a decreased likelihood of infection, suggesting that influenza vaccines designed to induce NA immunity alongside HA immunity could enhance protection against infection and reduce infectivity among vaccinated individuals upon infection [42].

The use of special diagnostic recombinants that do not show cross-reactivity in the HI test with immune sera from circulating influenza viruses eliminates distortion of the results of NA antibody detection [43]. Specially prepared viruses with irrelevant HA subtype H6 or H7 are commonly used for this purpose, which is reflected in our earlier works [19,44,45,46,47].

It is often believed that current IIVs do not elicit strong anti-NA responses [48], and that infection, but not vaccination, induces anti-NA antibodies in humans [13]. However, several studies, including our recent study, demonstrated that seasonal IIVs cause an increase not only in antibodies to HA but also in measurable response to NA, which lasted throughout the year after vaccination [19,49]. Our present study also found that seasonal IIVs induced immune responses to HA and NA of drifted influenza A/H1N1pdm09 and A/H3N2 strains (Figure 2) although seroprotection levels were lower (Figure 4). When assessing seroprotection, we used titers of NI antibodies 1:20. Despite extensive research on NI antibodies due to their broad cross-reactivity, specific seroprotection levels for NI antibodies have yet to be established. Recent studies have shown that pre-existing neuraminidase antibody titers of 1:40, as determined by ELISA, significantly reduced the duration of seasonal influenza A virus shedding in adults [10,42]. However, different studies have suggested that various NI antibody levels (ranging from 1:8 to ≥1:20) may confer protection against natural influenza infection [50,51,52]. It was observed that NA antibodies at protective levels, independent of HA antibodies, were less frequently induced, and no protective NI antibodies were formed to the A/H3N2 drift virus (Figure 5).

Our study also showed that individuals under 60 years of age had a lower response to drifted viruses following vaccination compared with those over 60 years old (Figure 2C and Figure 5). Statistically significant differences were only observed in the seroprotection levels against the NA of the A/H1N1pdm09 drift virus. This finding may be explained by the activation of memory B cells upon vaccination, with the breadth and quality of the response varying based on pre-existing immunity [53,54]. Previous studies have indicated a reduced immune response to the N1 protein of A/H1N1pdm09 viruses in middle-aged individuals [55]. However, it remains unclear whether early-life exposure to influenza viruses is crucial for the subsequent development of broadly cross-reactive NI antibodies, as observed for HA.

A reduced immune response to the N1 protein of drifted A/H1N1pdm09 virus was detected in individuals under 60 years old, and a very low content of protective antibodies against the N2 protein was observed in all age groups. These findings suggest that, in the case of a mismatch, vaccination may be necessary for individuals under 60 years of age, while the entire population may require vaccination against N2-containing viruses.

## 5. Conclusions

Seasonal vaccines lead to an increase in antibodies against both HA and NA of closely related and subsequently emerging influenza viruses. Drifted A/H1N1pdm09 and A/H3N2 viruses have shown significantly lower reactivity with NI antibodies compared with early influenza viruses. Following seasonal IIVs, seroprotection levels increased, including to drifted influenza A viruses and, to a greater extent, to A/H1N1pdm09. A diminished immune response to the N1 protein of drifted A/H1N1pdm09 virus was observed in subjects under 60 years of age. Based on our findings, it is hypothesized that in the cases of a HA mismatch, vaccination against N1-containing influenza viruses may be necessary for individuals under 60, while broader population-level vaccination against N2-containing viruses may be required.

The limitations of this study are related to the relatively small number of surveys. The study was conducted before the COVID-19 pandemic when quarantine measures and personal protective equipment were widely used. This changed the influenza situation; in particular, population immunity to influenza decreased.

## Figures and Tables

**Figure 1 vaccines-12-01334-f001:**
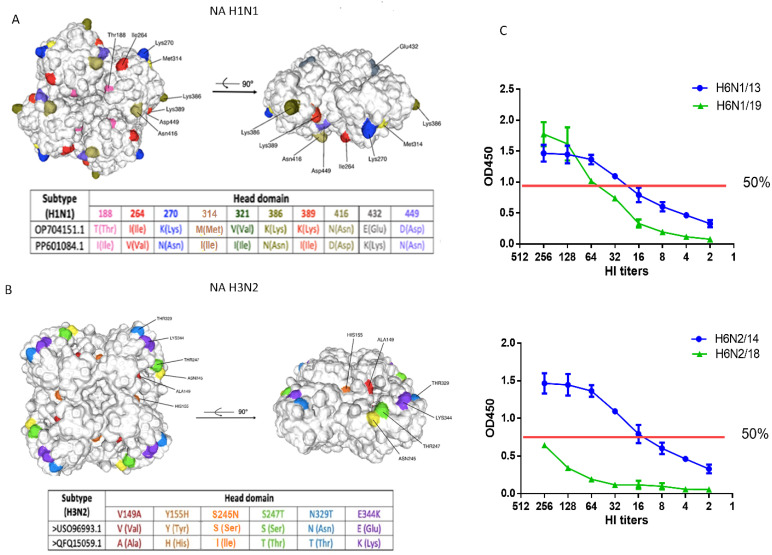
Molecular analysis and enzymatic activity of NA from A/H1N1pdm09 and A/H3N2 viruses. (**A**) A cartoon-style molecular model of NA from the A/Guangdong-Maonan/SWL1536/2019(H1N1)pdm09 virus, from amino acid 82 to 469 (numbering as in the case of the H1N1 2009 pandemic virus) with major amino acid substitutions in the structure indicated. (**B**) A molecular model of NA A/Brisbane/34/2018 (H3N2), from amino acid 82 to 469 (classical H3N2 strain numbering). (**C**) Enzymatic activity of NA of viruses A/H6N1 and A/H6N2 was studied in the desialyzation reaction of the high-molecular substrate (fetuin), sorbed on a polymeric carrier using peroxidase-labeled lectin. The OD450 was measured depending on the hemagglutinating activity of the viruses.

**Figure 2 vaccines-12-01334-f002:**
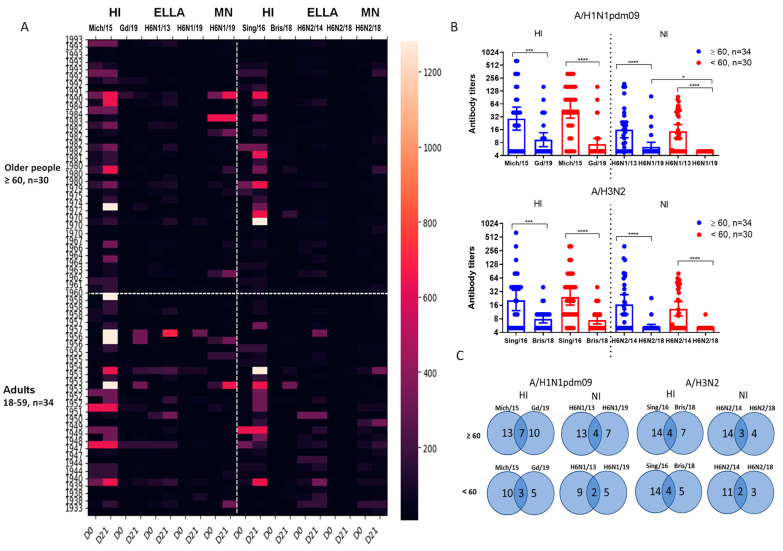
Antibodies to A/H1N1pdm09 and A/H3N2 viruses in sera of patients vaccinated with inactivated influenza vaccines (IIVs) corresponded to the WHO recommendations for the Northern Hemisphere during the 2018–2019 flu season. The total number of participants was 64: 30 people were 60 years old and older and 34 were under 60 years old. (**A**) Antibody levels to HA and NA of influenza viruses A/South Africa/3626/13 (H1N1)pdm09, A/Hong Kong/4801/2014 (H3N2), A/Guangdong-Maonan/SWL 1536/2019(H1N1)pdm09 and A/Brisbane/34/2018(H3N2) pre- and post-vaccination. (**B**) The HI and NI antibody levels to the vaccine and drifting viruses A/H1N1pdm09 and A/H3N2 in sera collected before vaccination. Each point represents an individual patient serum, here and below: *—*p* < 0.05, ***—*p* < 0.001, ****—*p* < 0.0001. (**C**) Combined seroconversions to the vaccine and drifted influenza viruses on day 21 following vaccination with IIVs regardless of the vaccine type, presented by Venn’s diagrams. Numbers in circles present the absolute number of responders to each virus. The total number of subjects was 64, with 30 people 60 years old or older and 34 under 60. The number of nonresponders to both antigens is not shown on Venn’s diagram. Seroconversion to influenza A virus antigens was defined as a fourfold increase in HI antibody titers and a twofold increase in NI antibody titers.

**Figure 3 vaccines-12-01334-f003:**
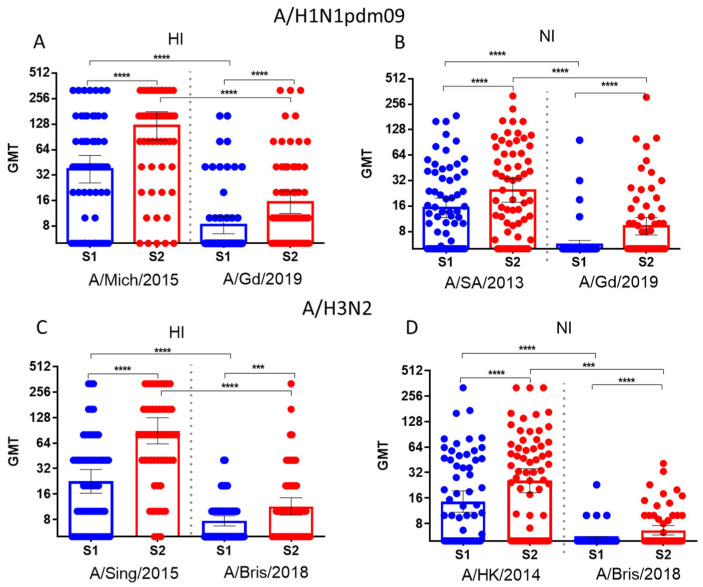
Antibody titers to HA and NA of influenza viruses in paired blood sera of patients vaccinated with seasonal IIVs 2018–2019 years of formulation (*n* = 64). S1—pre-vaccination antibody titers, S2—antibody titers 21 days after vaccination. The population in the analysis included all participants, regardless of age or vaccine type. Each dot represents an individual serum. (**A**) HI antibodies to A/Michigan/45/2015(H1N1)pdm09 and A/Guangdong-Maonan/SWL1536/2019(H1N1)pdm09 viruses. (**B**) NI antibodies to H6N1/13 and H6N1/19 influenza viruses. (**C**) HI antibodies to A/Singapore/INFIMH-16-0019/2016(H3N2) and A/Brisbane/34/2018(H3N2) virus. (**D**) NI antibodies to H6N2/14 and H6N2/18 influenza virus. ***—*p* < 0.001, ****—*p* < 0.0001.

**Figure 4 vaccines-12-01334-f004:**
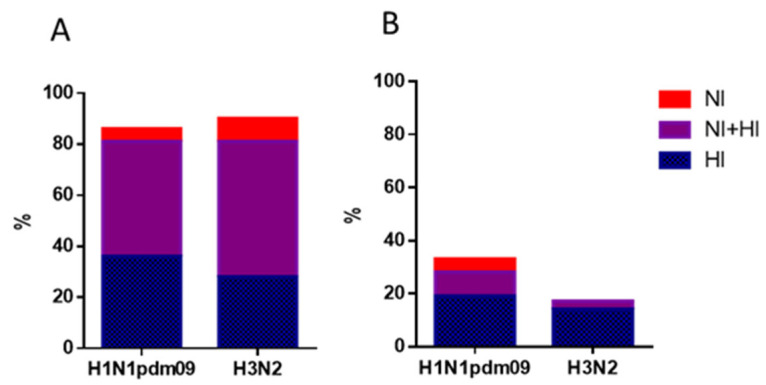
Proportions of individuals with antibody titers ≥ 1:40 for HI antibodies and ≥ 1:20 for NI antibodies to HA and NA of influenza viruses A/H1N1pdm09 and A/H3N2 post-vaccination with seasonal IIVs (*n* = 64). The population in the analysis included all participants, regardless of age or vaccine type. (**A**) The HI and NI antibodies to A/South Africa/3626/13 (H1N1)pdm09 and A/Hong Kong/4801/2014 (H3N2). (**B**) The HI and NI antibodies to A/Guangdong-Maonan/SWL 1536/2019(H1N1)pdm09 and A/Brisbane/34/2018 (H3N2).

**Figure 5 vaccines-12-01334-f005:**
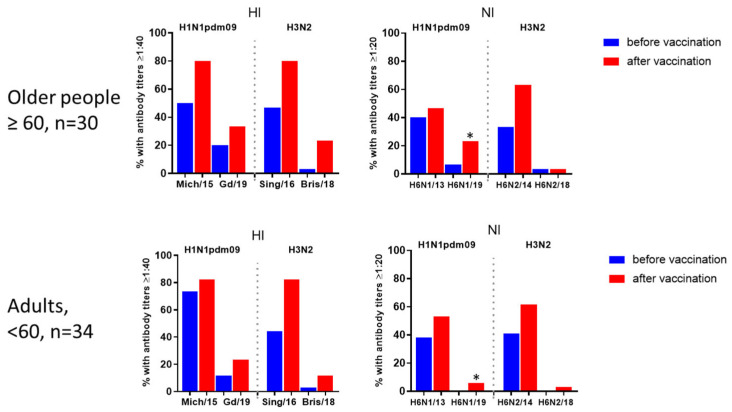
Seroprotection levels to HA and NA of influenza viruses A/H1N1pdm09 and A/H3N2 pre- and post-vaccination with seasonal IIVs in patients of different age groups. For HI antibodies, the seroprotection level was determined as 1:40, and for NI antibodies—as 1:20. *—*p* < 0.05, Fisher’s exact test.

**Figure 6 vaccines-12-01334-f006:**
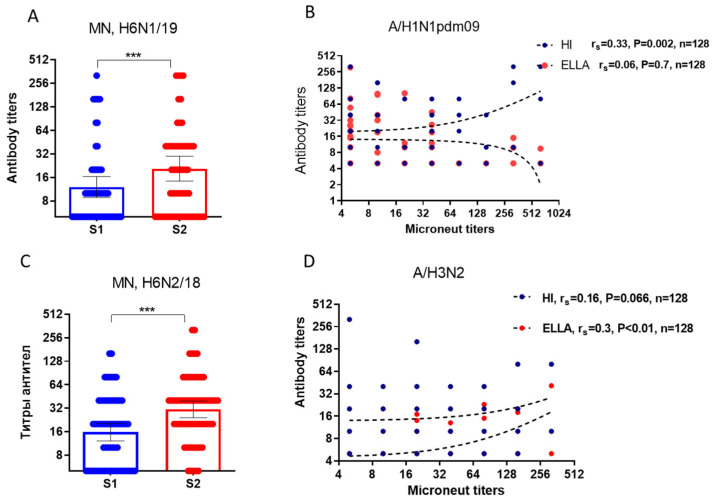
Results of the study of neutralizing antibodies using the MN test in MDCK cell line. S1—pre-vaccination antibody titers, S2—antibody titers 21 days after vaccination. ***—*p* < 0.001. (**A**) The NI antibodies to the H6N1/19 virus. The population in the analyses included all participants, regardless of age or vaccine type. (**B**) Correlation analysis of neutralizing antibodies and antibodies to HA and NA of the A/Guangdong-Maonan/SWL1536/2019(H1N1)pdm09 virus before and after vaccination. (**C**) The NI antibodies to the H6N2/18 virus. (**D**) Correlation analysis of neutralizing antibodies and antibodies to HA and NA of the A/Brisbane/34/2018(H3N2) virus before and after vaccination.

**Figure 7 vaccines-12-01334-f007:**
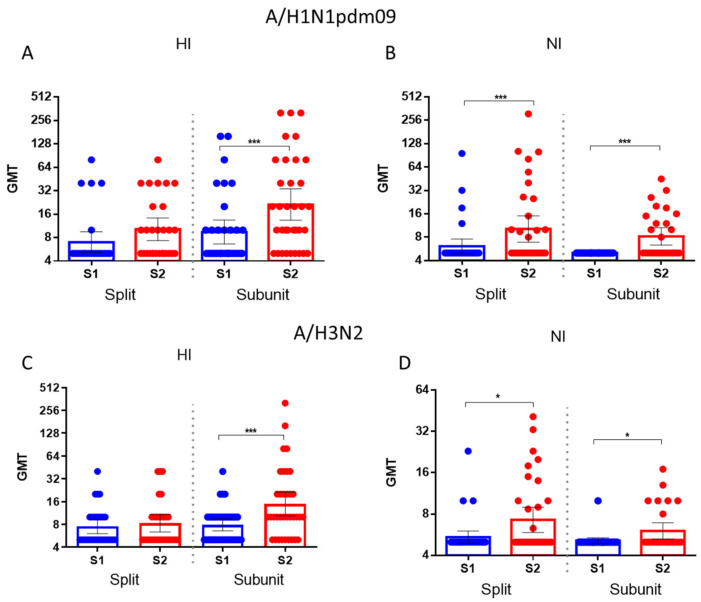
Antibody titers against drifted A/H1N1pdm09 and A/H3N2 HA and NA in patients vaccinated with split influenza vaccines (*n* = 29) and subunit influenza vaccines (*n* = 35). Each dot represents an individual serum. (**A**) HI antibodies to the A/Guangdong-Maonan/SWL1536/2019(H1N1)pdm09 virus. (**B**) NI antibodies to H6N1/19 influenza virus. (**C**) HI antibodies to A/Brisbane/34/2018(H3N2) virus. (**D**) NI antibodies to H6N2/18 influenza virus. *—*p* < 0.05, ***—*p* < 0.001.

**Table 1 vaccines-12-01334-t001:** Genome composition of diagnostic reassortant viruses of subtype H6 for use in ELLA.

Name	Origin of HA	Origin of NA
H6N1/13	A/Herring Gull/Sarma/51c/2006 (H6N1)	A/South Africa/3626/2013(H1N1)pdm09
H6N2/14	A/Herring Gull/Sarma/51c/2006 (H6N1)	A/HongKong/4801/2014 (H3N2)
H6N1/19	A/Herring Gull/Sarma/51c/2006 (H6N1)	A/Guangdong-Maonan/SWL 1536/2019(H1N1)pdm09
H6N2/18	A/Herring Gull/Sarma/51c/2006 (H6N1)	A/Brisbane/34/2018 (H3N2)

## Data Availability

The original contributions presented in the study are included in the article material; further inquiries can be directed to the corresponding authors.

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
