# Peer review of "Neuraminidase Antibody Response to Homologous and Drifted Influenza A Viruses After Immunization with Seasonal Influenza Vaccines"

_vaccines, 2024, doi:10.3390/vaccines12121334_

Round 1

Reviewer 1 Report

Comments and Suggestions for Authors

Very nice, useful study on the effect of the NA component in standard seasonal IIVs - but can the authors offer more information on the range of NA content in these seasonal vaccines - even approximate values?

Otherwise it is difficult to understand the dose-response relationship - that will be valuable in developing NA-based IIVs in future.

- from the plots presented, the serological responses look similar in strength to those induced by the known HA component?

Please also explain the need for the hybrid H6 virus experiments more clearly - it is not clear why these were necessary and is a bit confusing as these are non-human influenza viruses.

Author Response

First and foremost, the authors are very grateful to all reviewers for their assessment of our work and useful comments.

Point 1: Very nice, useful study on the effect of the NA component in standard seasonal IIVs - but can the authors offer more information on the range of NA content in these seasonal vaccines - even approximate values? Otherwise, it is difficult to understand the dose-response relationship - that will be valuable in developing NA-based IIVs in future.

Response 1. The authors thank the Reviewer for such a good assessment of our work. 

As stated by the vaccine manufacturers, the NA content in the vaccines was not determined. We added to Materials and Methods: “There is no specific information regarding the amount of NA in any of the vaccines used in our study. The inclusion of NA in the Grippol vaccine is mentioned in the product information, which can be found on the manufacturer's website (https://grippol.ru/en/grippol-kvadrivalent/instruction/, accessed on November 18, 2024”.

Point 2: From the plots presented, the serological responses look similar in strength to those induced by the known HA component?

Response 2. The authors thank the Reviewer for this remark. 

We added: “We did not compare levels of antibodies to HA and NA because these values were obtained by different methods. However, Figure 3 shows that the mean titers of HI antibodies were almost always about two times lower than those of NI antibodies. Given that there is no precise definition for protective titers for NI antibodies, the clinical significance of these values is still unclear“.

Point 3: Please also explain the need for the hybrid H6 virus experiments more clearly - it is not clear why these were necessary and is a bit confusing as these are non-human influenza viruses.

Response 3. The authors thank the Reviewer for this comment. We have added to the text of the paper:

“The use of special diagnostic recombinants that do not show cross-reactivity in the HI test with immune sera from circulating influenza viruses eliminates distortion of the results of NA antibody detection [43]. Specially prepared viruses with irrelevant HA subtype H6 or H7 are commonly used for this purpose which is reflected in our earlier works” [19, 44, 45, 46, 47].

Reviewer 2 Report

Comments and Suggestions for Authors

The study investigated the immune response to neuraminidase (NA) and hemagglutinin (HA) in influenza A viruses following vaccination with trivalent inactivated influenza vaccines (IIVs). The authors found that while seasonal IIVs led to increased antibody production for both homologous and drifted influenza strains, antibody levels, especially against drifted viruses, were generally higher in older patients. The study also noted that NA content in vaccines is unregulated, even though NA antibodies have shown potential in reducing infection severity and aiding cross-protection when HA-targeted immunity is mismatched. Overall, this study was designed well and data are reliable, while the abstract is complex and not clear, author should re-write it. Othe concerns are listed as below:

Repetitive sentences in the Ms, such as in lines 49-51, authors should re-write them. In addition, a lot of grammar errors throughout this Ms, such as in line 54: “it is worth take into account immunity directed against……”. Here, I will not point out each one individually.

For the sentence between lines 55 and 57: “when studying susceptibility after vaccination and consider NA as a critical target in future promising influenza vaccine platforms”, I can't precisely grasp the meaning the author is trying to convey. Pls re-write this sentence or give a more detailed explanation.

Lack of Citations: The article frequently lacks references for significant claims, especially when discussing background and comparisons to past studies. Authors should add references to support for these statements.

Unclear Labeling: For better clarity in Figure 3, authors should have a more detailed legend to explain "S1" and "S2".

Comments on the Quality of English Language

English writing needs to be improved

Author Response

First and foremost, the authors are very grateful to all reviewers for their assessment of our work and useful comments.

The study investigated the immune response to neuraminidase (NA) and hemagglutinin (HA) in influenza A viruses following vaccination with trivalent inactivated influenza vaccines (IIVs). The authors found that while seasonal IIVs led to increased antibody production for both homologous and drifted influenza strains, antibody levels, especially against drifted viruses, were generally higher in older patients. The study also noted that NA content in vaccines is unregulated, even though NA antibodies have shown potential in reducing infection severity and aiding cross-protection when HA-targeted immunity is mismatched. Overall, this study was designed well and data are reliable, while the abstract is complex and not clear, author should re-write it. Other concerns are listed as below:

Response. The authors thank the Reviewer for this remark.  We appreciate a work done by the Reviewer and are thankful for the criticism and valuable comments. Abstract was rephrased for clarity. Answers to other comments, please, find below.

Point 1: Repetitive sentences in the Ms, such as in lines 49-51, authors should re-write them. In addition, a lot of grammar errors throughout this Ms, such as in line 54: “it is worth take into account immunity directed against……”. Here, I will not point out each one individually.

Response 1. We thank the Reviewer for criticism. Repetitive sentence was removed. English was corrected by professional English translator

Point 2: For the sentence between lines 55 and 57: “when studying susceptibility after vaccination and consider NA as a critical target in future promising influenza vaccine platforms”, I can't precisely grasp the meaning the author is trying to convey. Pls re-write this sentence or give a more detailed explanation.

Response 2. We agree with the Reviewer that the sentence was poorly worded. We have rephrased it as follows: “In case of genetic drift or antigenic shift in circulating strains, vaccine-induced HI antibodies might be less effective at cross-neutralization. Then immunity directed against NA – the second most important antigenic component of the influenza virus - comes to the fore. This allows NA to be considered as a critical target in future promising influenza vaccine platform [4, 5].”

Point 3: Lack of Citations: The article frequently lacks references for significant claims, especially when discussing background and comparisons to past studies. Authors should add references to support for these statements.

Response 3. The authors thank the Reviewer for this remark.  We have added references.

Point 4: Unclear Labeling: For better clarity in Figure 3, authors should have a more detailed legend to explain "S1" and "S2".

Response 4. The authors thank the Reviewer for this remark.  We have added an explanation to the figure legend and in the text.

Reviewer 3 Report

Comments and Suggestions for Authors

Humoral immunity directed at neuraminidase (NA) of the influenza virus may soften the severity of infection caused by the new antigenic variants of influenza viruses. Evaluation of NA-inhibiting (NI) antibodies together with antibodies to hemagglutinin (HA) may enhance research of antibody response to influenza vaccines. This study examined 64 pairs of sera from patients vaccinated in 2018 with seasonal inactivated trivalent influenza vaccines (IIVs) with the formula recommended by WHO for the 2018-2019 influenza season. They used ELLA to evaluate NI and HI antibody levels to drift influenza viruses A/Guangdong-Maonan/SWL1536/2019(H1N1)pdm09 and A/Brisbane/34/2018(H3N2) before vaccination and 21 days after vaccination. They also performed microneutralization (MN) test in the MDCK cell line with A/H6N1 and A/H6N2 viruses.  They found that seasonal IIVs induce a substantial immune response of NI antibodies to influenza A/H1N1pdm09 and 23 A/H3N2 viruses. They also found a significantly reduced ‘herd’ immunity to drift influenza viruses A/H1N1pdm09 24 and A/H3N2, compared to previously circulating strains. The conclude that the Drift A/H1N1pdm09 and A/H3N2 viruses demonstrated significantly lower reactivity with NI antibodies to early influenza viruses. They hypothesized that in the case of mismatched HA, it will most likely be necessary to vaccinate against the N1-containing influenza virus those who under 60 years of age, 36 and the entire population against virus possessing N2. These findings are very interesting. However, I have the following comments and suggestions that need to be addressed before publication.

1.      Line 17-21. Antibodies to drift influenza viruses ….. Please rephrase this sentence.

2.      Line 33: NI and NI antibodies? Please clarify.

3.      Line NA.NA. add space.

4.      Line 145: What is PB? What is the pH value?

5.      Lines 149-150: 0.01M FB? Is this correct?

6.      Lines 159-161: Please provide more information about the HI test.

7.      Lines 152-153: What OD? Which dye was used?

8.      Line 162: Please provide reference(s) to this method.

9.      Lines 214-215: Isoleucine to valine (V321I)? Please correct it.

10.  Lines 288-290: Please describe in more detail the results shown in Figure 3.

11.  Line 301: Figure 4 and 5 should be Figures 4 and 5.

12.  Figure 6: It was claimed that n=64, yet only 19 spots in 6B and 16 spots in 6D. Please explain.

Comments on the Quality of English Language

 Line 17-21. Antibodies to drift influenza viruses ….. Please rephrase this sentence.

Line 301: Figure 4 and 5 should be Figures 4 and 5.

Author Response

First and foremost, the authors are very grateful to all reviewers for their assessment of our work and useful comments.

Point 1: Line 17-21. Antibodies to drift influenza viruses … Please rephrase this sentence.

Response 1.  We thank the Reviewer for criticism. The sentence “Antibodies to drift influenza viruses A/Guangdong-Maonan/SWL1536/2019(H1N1)pdm09 and A A/Brisbane/34/2018(H3N2) were studied before vaccination and 21 days after vaccination” was replaced with “Antibodies to antigenically drifted influenza A/Guangdong-Maonan/SWL1536/2019(H1N1)pdm09 and A/Brisbane/34/2018(H3N2) were studied before vaccination and 21 days after vaccination.”

Point 2: Line 33: NI and NI antibodies? Please clarify.

Response 2. The authors thank the Reviewer for this remark. Corrected.

Point 3: Line 77. NA.NA. add space.

Response 3. The authors thank the Reviewer for this remark. Corrected.

Point 4: Line 145: What is PB? What is the pH value?

Response 4. We are sorry for typing mistake. We meant phosphate-buffered saline (PBS), pH ~ 7.4. Text was corrected accordingly.

Point 5: Lines 149-150: 0.01M FB? Is this correct?

Response 5. We are sorry for typing mistake. We meant 0.01 M PBS. Text was corrected accordingly.

 Point 6: Lines 159-161: Please provide more information about the HI test.

Response 6. As required, the description of the HI assay was presented in more detail.

The following paragraph was added: “In brief, Influenza antigens bearing the same hemagglutinin as in the vaccine strains were used. Sera were treated with receptor-destroying enzyme (RDE, Denka Seiken Co., Tokyo, Japan) according to the manufacturer’s instructions. Each blood serum sample was 2-fold serially diluted in 96-well polypropylene U-bottom plates starting from 1:10 and mixed with 4 HAU of influenza antigen. After 1 h of incubation, 0.5% suspension of hens red blood cells was added. Antibody seroconversion was identified as a four-fold or greater increase in the HI antibody titer compared to baseline.”

Point 7: Lines 152-153: What OD? Which dye was used?

Response 7.. TMB was used as a dye. Optical density (450 nm) was measured by microplate reader. The corresponding corrections have been made to the text.

Point 8: Line 162: Please provide reference(s) to this method.

Response 8. We thank the Reviewer for this comment.  We have added a link to the electronic document:

(World Health Organization. Manual for the Laboratory Diagnosis and Virological Surveillance of Influenza. 2011. Available online: https://apps.who.int/iris/handle/10665/44518 (accessed on 13 November 2024).

Point 9: Lines 214-215: Isoleucine to valine (V321I)? Please correct it.

Response 9. We thank the Reviewer for this comment. “Structural analysis showed that the amino acid change from isoleucine to valine (V321I)…” was replaced with “Structural analysis showed that the amino acid change from valine to isoleucine (V321I)…”

Point 10: Lines 288-290: Please describe in more detail the results shown in Figure 3.

Response 10. The authors thank the Reviewer for this comment.  We have supplemented the description of Figure 3

Point 11: Line 301: Figure 4 and 5 should be Figures 4 and 5.

Response 11. We thank the Reviewer for criticism. “Figure 4 and 5” was replaced with “Figures 4 and 5.”

Point 12: Figure 6: It was claimed that n=64, yet only 19 spots in 6B and 16 spots in 6D. Please explain.

Response 12. The fact that the diagram has fewer points than the number of bases indicated on the diagram is explained by the fact that antibody titers often have similar values ​​and the program arranges them so that only one point is visible.

 COMMENTS ON THE QUALITY OF ENGLISH LANGUAGE

Point 13: Line 17-21. Antibodies to drift influenza viruses … Please rephrase this sentence.

Response 13. We thank the Reviewer for criticism. The sentence “Antibodies to drift influenza viruses A/Guangdong-Maonan/SWL1536/2019(H1N1)pdm09 and A A/Brisbane/34/2018(H3N2) were studied before vaccination and 21 days after vaccination” was replaced with “Antibodies to antigenically drifted influenza A/Guangdong-Maonan/SWL1536/2019(H1N1)pdm09 and A/Brisbane/34/2018(H3N2) were studied before vaccination and 21 days after vaccination.”

Point 14: Line 301: Figure 4 and 5 should be Figures 4 and 5.

Response 14. We thank the Reviewer for criticism. “Figure 4 and 5” was replaced with “Figures 4 and 5.”

English language was corrected by professional translator.

Reviewer 4 Report

Comments and Suggestions for Authors

The manuscript details a series of studies on the efficacy of influenza vaccines in cases of incompatibility between the vaccine and the epidemic strains. The authors' clear and precise objective was to evaluate the cross-reactivity and functional properties of inhibitory antibodies (before vaccination and 21 days after) of viral neuraminidase produced as a result of immunization with vaccines against seasonal influenza (A/H1N1pdm09 and A/H3N2). The manuscript's significant contribution to the field was well conducted, with objectives that guide the reader and a discussion that robustly supports the results. The figures are of good quality, and the references are essential for understanding the text. The hypothesis that individuals under 60 should be vaccinated against N1 and the entire population with N2 is pertinent. We strongly recommend its publication. However, we suggest a general review of the English and an improvement of the abstract to summarize the manuscript's crucial findings better.

Author Response

First and foremost, the authors are very grateful to all reviewers for their assessment of our work and useful comments.

The manuscript details a series of studies on the efficacy of influenza vaccines in cases of incompatibility between the vaccine and the epidemic strains. The authors' clear and precise objective was to evaluate the cross-reactivity and functional properties of inhibitory antibodies (before vaccination and 21 days after) of viral neuraminidase produced as a result of immunization with vaccines against seasonal influenza (A/H1N1pdm09 and A/H3N2). The manuscript's significant contribution to the field was well conducted, with objectives that guide the reader and a discussion that robustly supports the results. The figures are of good quality, and the references are essential for understanding the text. The hypothesis that individuals under 60 should be vaccinated against N1 and the entire population with N2 is pertinent. We strongly recommend its publication.

Point 1: However, we suggest a general review of the English and an improvement of the abstract to summarize the manuscript's crucial findings better.

Response 1. We thank the Reviewer for the positive assessment of our work and valuable comments. The Abstract was rephrased for clarity. English language was corrected by professional translator.

Round 2

Reviewer 2 Report

Comments and Suggestions for Authors

I am ok with the new version.

Comments on the Quality of English Language

a lot of improvement in the new verson

Reviewer 3 Report

Comments and Suggestions for Authors

The authors properly addressed all my comments and suggestions. The revised version is significantly improved.